# An Unsupervised Mutual Information Feature Selection Method Based on SVM for Main Transformer Condition Diagnosis in Nuclear Power Plants

**Wenmin Yu** [1] , **Ren Yu** [1,*] **and Jun Tao** [2]

1   School of Nuclear Science and Techniques, Naval University of Engineering, Wuhan 430034, China; hust_ywm2012@163.com
2   China Nuclear Power Operation Management Co., Ltd., Jiaxing 314300, China; taoj@cnnp.com.cn
*   Correspondence: 18071068480@163.com

**Abstract:** Dissolved gas in oil (DGA) is a common means of monitoring the condition of an oil-immersed transformer. The concentration of dissolved gas and the ratio of different gases are important indexes to judge the condition of power transformers. Monitoring devices for dissolved gas in oil are widely installed in main transformers, but there are few recorded fault data of main transformers. The special operation and maintenance modes of main transformers leads to the fault modes particularity of main transformers. In order to solve the problem of insufficient samples and the feature uncertainty, this paper puts forward an unsupervised mutual information method to select the feature verified by the optimized support vector machine (SVM) model of particle swarm optimization (PSO) method and tries to find the feature sequence with better performance. The methos is validated by data from nuclear power transformers.

**Keywords:** main transformer; condition monitoring; unsupervised mutual information; feature selection; DGA

## 1. Introduction

Power transformers that work under harsh environments would experience thermal decomposition of oil and cellulose insulation materials, such as arcing, corona discharge, low energy sparks, severe overloading, overheating of insulation systems and pump motor failures. These conditions alone or in combination can produce combustible and noncombustible gases [1] Detection of anomalies requires an assessment of the amount of gas produced. Gas in oil-immersed transformers can be used to identify fault types, including thermal and electrical interference. Gases obtained from chromatographic analysis of insulating oils may contain dissolved carbon monoxide ($CO$), carbon dioxide ($CO_2$), nitrogen ($N_2$), hydrogen ($H_2$), methane ($CH_4$), acetylene ($C_2H_2$), ethylene ($C_2H_4$), and ethane ($C_2H_6$). The composition, formation rate and specific content ratio of dissolved gas can be used to indicate transformer condition.

The composition and content of dissolved gases in oil of transformer insulation can reflect the operation condition of transformer to a great extent thus dissolved gas analysis (DGA) has become an effective method for fault diagnosis of oil-immersed transformers [2].

Organizations such as the Institute of Electrical and Electronics Engineers (IEEE) and the International Electrotechnical Commission (IEC) recommend a variety of diagnostic techniques [3], depending on the type of transformer and operating conditions. Some of the most commonly used techniques include Doernenburg ratio, Rogers ratio, Duval triangle model, etc. These classical diagnostic methods mostly take the ratio of different gases as the characteristic input and then judge the actual operating condition of the transformer by the threshold value formed by experience or statistical methods. Fuzzy network, support vector machine, artificial neural network, and other commonly used artificial intelligence methods

are also generally introduced into the field of power transformer fault diagnosis [4–7]. However, in the studies of different scholars, the features used as the basis of intelligent diagnosis are often different.

In previous studies, in addition to problems in diagnostic methods, there are the following phenomena in monitoring data: less test data, less available data sets and unbalanced data type distribution, which bring great problems to algorithm verification [8].

Main transformers are important equipment for power generation of nuclear facilities. They are in a high-load long-term condition and are more prone to failure caused by aging [9]. Meanwhile, due to the particularity of nuclear power refueling overhaul and the conservative culture of nuclear power [10], the maintenance strategy of nuclear equipment is more rigorous and conservative, and the failure modes of main transformers may be slightly different. The data of the main transformers are classified separately in the IEC database [2], which shows transformer performance difference in nuclear industry.

Due to the particularity of nuclear power transformers, there are less marked data and more constraints on the monitoring data that can be used for research. The features are important inputs of the diagnostic algorithm. High-dimensional features bring high computational cost and the risk of "over-fitting". Dimensionality reduction or selection is an important research direction.

In this paper, an unsupervised mutual information feature algorithm is proposed for feature selection of different features proposed in the current classical algorithm and intelligent algorithm, as a pattern recognition method, SVM can construct the optimal classification hyperplane under the condition of small sample learning and distinguish transformer conditions according to the input features. The main transformer condition diagnosis model based on support vector machine is adopted for diagnosis in this paper, and the case data of main transformer is verified.

## 2. Framework of the Feature Selection Method Based on the SVM Model for Main Transformers

The research framework for the feature selection method based on the architecture of the SVM model for the main transformers is shown in Figure 1.

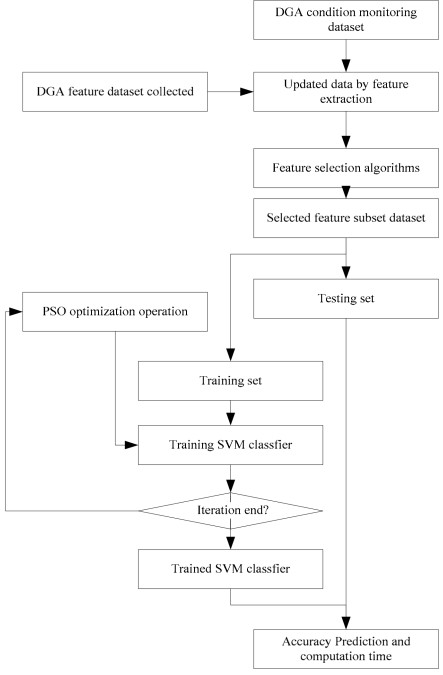

**Figure 1.** Framework of Feature Selection method based on SVM model for main transformers.

The gas concentration values measured in the continuous operation process of nuclear power transformer are obtained, and there is almost no-fault data.

The features used in various power transformer condition diagnosis methods based on dissolved gas in oil are extensively studied. On this basis, the initial feature set is formed.

The unsupervised mutual information feature extraction algorithm is adopted to extract features, and the set of sequence features is obtained according to the weight coefficient from high to low.

In the feature set, different number of feature sets are selected sequentially and verified by optimized SVM model for transformer fault diagnosis.

In order to reduce the contingency of the experiment, the 5-fold verification method is used to process the training samples and test samples to verify the validity of the selection feature in the diagnosis of the nuclear power transformer condition diagnosis.

Based on the accuracy of diagnosis, the feasibility of different feature extraction algorithms in the condition diagnosis of main transformers is analyzed.

## 3. Condition Diagnosis Model for Main Transformer

Condition diagnosis model is important to verify the feature selection algorithm and to determine the accuracy and rapidity of transformer condition diagnosis. SVM is a machine learning method based on statistical learning theory, compared with other algorithms, it can well solve practical problems such as small sample, nonlinear [11], the PSO algorithm can converge fast in the parameter optimization. The SVM optimized with PSO in the field of power transformer fault diagnosis has further application [12].

### 3.1. Support Vector Machine

State diagnosis of main transformers, As the case of a typical nonlinear classification problem, the overall plan of SVM is the first use of a nonlinear transform the input space data is mapped to a high-dimensional feature vector space, and then in the feature space of the optimal separating hyperplane is constructed, linear classification, after the last map back to the original space Became a nonlinear classification of input space [13].

SVM settings

At present, the commonly used kernel functions are mainly polynomial kernel function, radial basis (RBF) kernel function, hyperbolic tangent (sigmoid) kernel function, and so on. This paper mainly uses the RBF kernel function to the apply to SVM model.

### 3.2. PSO for Optimal Parameters

PSO is a kind of evolutionary computation, the basic idea of which is to find the optimal solution through the cooperation and information sharing between individuals in the group. It mimics a bird in a flock by designing a massless particle with just two properties: speed, which represents how fast it is moving, and position, which represents the direction it is moving. Each particle separately searches for the optimal solution in the search space, and records it as the current individual extreme value, and shares the individual extreme value with other particles in the whole particle swarm and finds the optimal individual extreme value as the current global optimal solution of the whole particle swarm. All particles in a swarm adjust their speed and position based on the current individual extremum they find, and the current global optimal solution shared by the whole swarm [14].

PSO-SVM Parameter Settings

$C_1$: the initial value is 1.5, local search capability of PSO parameters

$C_2$: 1.7 initially, PSO parameter global search capability

Maxgen: The initial value is 200, the maximum number of evolutions

Sizepop: the initial value is 20 and the maximum size of the population

K: initial 0.6 (k belongs to [0.1, 1.0]), the relationship between the speed and x ($V = KX$)

WV: The initial value is 1 (wV best belongs to [0.8, 1.2]), and the rate updates the elasti coefficient before the speed in the formula

WP: The initial value is 1, the elastic coefficient in front of the velocity in the population renewal formula

V: Initial 5, SVM Cross Validation parameter

Popcmax: the maximum value of the change in the SVM parameter C, initially 100.

Popcmin: the initial value is 0.1, the minimum change of SVM parameter C.

Popgmax: the initial value is 1000, the maximum value of the change of THE SVM parameter G.

Popgmin: the initial value is 0.01, the minimum change value of the SVM parameter C.

## 4. Feature Selection Algorithms for Main Transformer Condition

This chapter introduces the unsupervised mutual information filtering feature sorting method used in feature selection. In feature selection, the relevance of each feature is calculated first, the importance of the feature is evaluated by the forward sequential search, and finally an ordered feature sequence is output.

### 4.1. Stepwise Feature Selection Process

The process of stepwise feature selection is to select a feature from the unselected feature set each time and add the feature set S. In accordance with the selection order, the feature set outputs an ordered feature sequence.

When initializing, the feature set is empty. The unselected feature set is the complete set of all known features.

After each step selection, the feature set increases the feature set selected in this step, while the feature set not selected reduces the feature set selected in this step. Until the unselected feature set is empty.

### 4.2. Selection Principle

The principle of "minimum redundancy—maximum correlation" which is similar to the famous supervised feature selection method is adopted [15], and the selection of the mth feature is based on:

$$l_m = arg\ max_{f_i \in U_m}\{Rel(f_i) - \frac{1}{m-1}\sum_{f_t \in S_{m-1}} Red(f_i, f_t)\} \tag{1}$$

where $U_m$ represents the set of unselected features in the current step

$f_i$ represents a feature in the unselected feature set in the current step;

$Rel(f_i)$ represents Relevance of feature $f_i$, which is the average mutual information between feature $f_i$ and any other one in the whole feature set is defined as $Rel(f_i)$. $Rel(f_i)$ can be calculated with Formula (2).

$$Rel(f_i) = \frac{1}{n}\sum_{t=1}^{n} I(f_i; f_t) = \frac{1}{n}(H(f_i) + \sum_{1 \leq t \leq n, t \neq i} I(f_i; f_t)) \tag{2}$$

$S_{m-1}$ is the selected feature set in the current step;

$Red(f_i, f_t)$ is the redundancy of feature $f_i$ relative to selected feature $f_t$. $Red(f_i, f)_t$ can be calculated with Formula (3).

$$Red(f_i, f_t) = Rel(f_t) - Rel(f_t|f_i) \tag{3}$$

$Rel(f_t|f_i)$ is conditional relevance of $f_t$ with $f_i$, $Rel(f_t|f_i)$ can be calculated with Formula (4).

$$Rel(f_t|f_i) = \frac{H(f_t|f_i)}{H(f_t)} \times Rel(f_t) \tag{4}$$

*4.3. Relationship with Supervised Algorithms*

When the data type is supervised, the labels of the class can represent the information of the whole feature set.

Then relevance of feature $f_i$ can be defined as

$$Rel(f_i) = I(f_i, c) \tag{5}$$

where $c$ in Formula (5) is the class label [16].

Redundancy between feature $f_i$ and the selected feature $f_t$ is defined as

$$Red(f_i, f_t) = I(f_i, f_t) \tag{6}$$

According to the principle of mathematics [17], relevance in an unsupervised algorithm is the lower bound of relevance in a supervised algorithm, and redundancy in an unsupervised algorithm is proportional to the redundancy in a supervised algorithm. When the initial feature set is approximately equal to the labels of the class, the sequence features obtained by the unsupervised algorithm are highly correlated with the sequence features obtained by the supervised algorithm.

## 5. Experiment and Validation

*5.1. Experiment Description*

The internal fault mode of power transformer is mainly mechanical fault, thermal fault and electrical fault, the latter two types of faults is the major issues, and mechanical fault is often shown in the form of thermal fault or electrical fault [18]. General power transformers are often subdivided into fault modes according to the degree of heating or arcing.

Due to safety culture of nuclear power plants, maintenance strategy for the main transformers tends to be conservative and strict; The failure of nuclear power transformers is rare to happen, and the failure data that can be accessed to publicly is very few. Available fault data cannot cover all the modes. Therefore, the condition of the main transformers are divided into the following three types in this paper, and only two summative failure modes are reserved and the corresponding as illustrated in Table 1.

**Table 1.** Code of power transformer operation condition.

| NO. | FAILURE MODE | CONDITION TYPE | CODE |
|:---:|:---:|:---:|:---:|
| 1 | Partial discharge | | |
| 2 | Low-energy discharge | Electrical fault | 1 |
| 3 | High-energy discharge | | |
| 4 | Thermal fault < 300 °C | | |
| 5 | Thermal fault 300 to 700 °C | Thermal fault | 2 |
| 6 | Thermal fault > 700 °C | | |
| 7 | | Normal | 3 |

The fault data used in the experiment in this paper are reactor-related transformer data obtained from IEC TC database, and the feature selection data and normal data are the monitoring values of a nuclear power main transformer under normal operation. The data can be obtained in the Supplementary Material. A nuclear power plant generator, 24 kV voltage, is stepped up to 500 kV and connected to the 500 KV power grid through the main transformer. The main transformer is a three single-phase transformer, each phase capacity 410 MVA. The neutral points on the high voltage side are connected and directly grounded. Oil is regularly sampled and analyzed once every 3 months manually. Sampling intervals can sometimes be uneven, depending on special focus judged by sampling staff or adjustment by work schedules.

5.1.1. Sample Data

Gas concentration of DGA method is analyzed in one part per million (PPM). To facilitate the presentation and analysis of the characteristics in the figure, logarithmic processing is performed for each monitoring value in Figures 2 and 3. The box diagram of the monitored data value in the sample data is shown in Figure 2. Sample type distribution and monitoring value distribution of each basic feature are presented in Figure 3.

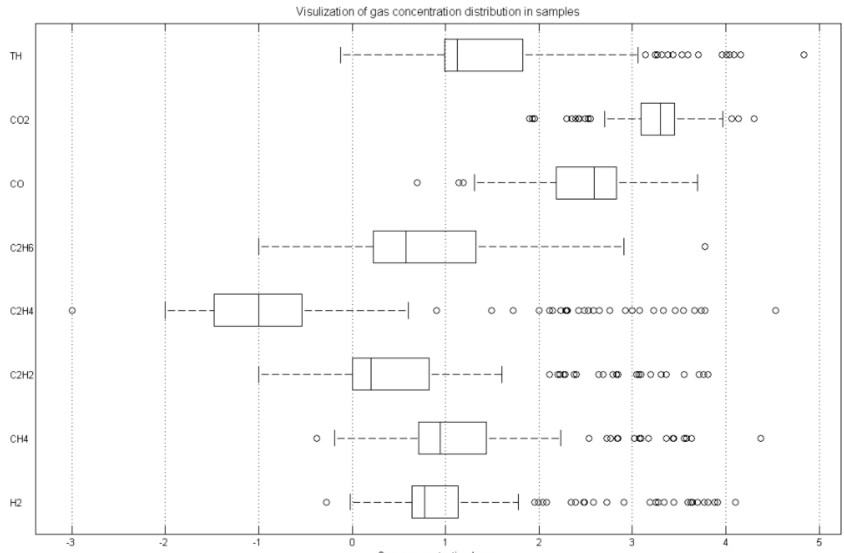

**Figure 2.** Box diagram of the monitored data value in the sample data.

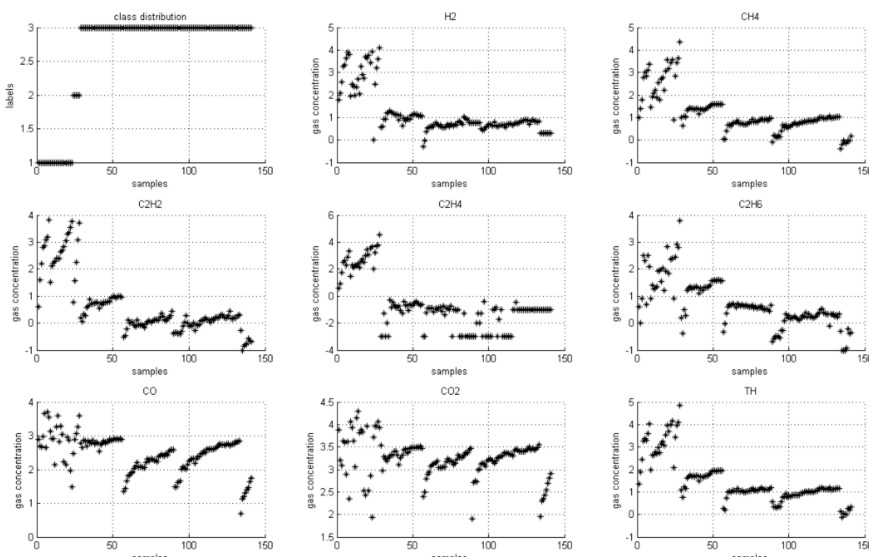

**Figure 3.** Sample type distribution and monitoring value distribution of each basic feature.

5.1.2. Basic Features by DGA Condition Monitoring

Typical gases measured in the DGA method of the main transformers include $H_2$, $CH_4$, $C_2H_2$, $C_2H_4$, $C_2H_6$, $CO$, $CO_2$ as illustrated in Figures 2 and 3.

$$TH = CH_4 + C_2H_2 + C_2H_4 + C_2H_6$$

Other features commonly used in transformer condition diagnosis methods are shown in Table 2.

**Table 2.** Common features of transformer DGA condition diagnosis methods.

| NO. | FEATURE | DIAGNOSIS METHOD |
|---|---|---|
| 1 | $\%C_2H_2$ | |
| 2 | $\%C_2H_4$ | The Duval Triangle method [2] |
| 3 | $\%CH_4$ | |
| 4 | $CH_4/H_2$ | |
| 5 | $C_2H_4/C_2H_6$ | The three basic gas ratios of IEC 599/IEC 60599 [19] |
| 6 | $C_2H_2/C_2H_4$ | |
| 7 | $C_2H_2/CH_4$ | Doernenberg Ratios [19] |
| 8 | $C_2H_6/C_2H_2$ | |
| 9 | $C_2H_2/H_2$ | Two new Gas ratios in IEC 60599 |
| 10 | $CO_2/CO$ | |
| 11 | $\varphi(H_2)$ | |
| 12 | $\varphi(CH_4)$ | |
| 13 | $\varphi(C_2H_6)$ | Other approach [20] |
| 14 | $\varphi(C_2H_4)$ | |
| 15 | $\varphi(C_2H_2)$ | |
| 16 | $\%H_2$ | fourth % ratio [21] |

The symbols in the Duval triangle method shown in Table 1 are denoted as $\%C_2H_2 = 100x/(x+y+z)$; $\%C_2H_4 = 100y/(x+y+z)$; $\%CH_4 = 100z/(x+y+z)$; with $x = (C_2H_2)$; $y = (C_2H_4)$; $z = (CH_4)$ in PPM.

$\varphi(H_2)$, $\varphi(CH_4)$, $\varphi(C_2H_6)$, $\varphi(C_2H_4)$ and $\varphi(C_2H_2)$ in Table 1 represent the contents of five characteristic gases, respectively, and Total Combustion Gases (TCG) as in: TCG = $H_2 + CH_4 + +C_2H_4 + C_2H_6 + C_2H_2$; $\varphi(H_2) = H_2/TCG$; $\varphi(CH_4) = CH_4/TCG$; $\varphi(C_2H_6) = C_2H_6/TCG$; $\varphi(C_2H_4) = C_2H_4/TCG$; $\varphi(C_2H_2) = C_2H_2/TCG$;

And $\%H_2 = 100 * H_2/(H_2 + C_2H_6 + CO + CO_2)$.

As shown in Figures 2 and 3, the class of the sample is not balanced and several basic features are similar in distribution. Unsupervised feature extraction is adopted to obtain the feature set with maximum correlation and minimum redundancy.

### 5.2. Extracted Feature Sequence by Unsupervised MI V.S. Supervised MI

Table 3 shows the comparison of feature sequences extracted by the unsupervised mutual information method and supervised mutual information method and their corresponding weights. As obtained from Table 3.

Table 3 Extracted feature sequence by unsupervised MI V.S. supervised MI the mutual information of unsupervised feature extraction method selects the same first feature from cases of nuclear power transformer DGA with that of supervised method, while the weight value, generated by supervised feature extraction algorithm is greater than the corresponding values in the unsupervised algorithm, which means supervised method has more dynamic to choose the first feature; In other steps, selected feature is not the same, but the unsupervised algorithm has a strong dynamic at each step.

### 5.3. Results of the Diagnosis by Optimized SVM

5.3.1. Diagnostic Precision with Supervised Mutual Information Feature Selection Method versus Unsupervised Approach

According to the framework described in Section 2, a PSO-optimized SVM model is applied to classify cases to reflect the fitness of the selected features.

The feature sequences selected by the supervised mutual information feature selection algorithm and the unsupervised mutual information feature selection algorithm are applied, respectively. Different numbers of features are selected from feature sequences obtained by both methods, and the diagnostic accuracy of the both method is shown in Figure 4, in which the red point is where the best fitness is obtained.

**Table 3.** Extracted feature sequence by unsupervised MI V.S. supervised MI.

| No. | Supervised MI | | Unsupervised MI | |
|---|---|---|---|---|
| | Feature | Weight | Feature | Weight |
| 1 | $\%C_2H_4$ | 0.380665 | $\%C_2H_4$ | 0.31706 |
| 2 | $\%C_2H_2$ | 0.021415 | $\%H_2$ | 0.276004 |
| 3 | $\varphi(C_2H_4)$ | 0.095868 | $\varphi(C_2H_2)$ | 0.302481 |
| 4 | $C_2H_4/C_2H_6$ | 0.052677 | $\varphi(C_2H_4)$ | 0.306476 |
| 5 | $C_2H_2/CH_4$ | 0.058468 | $\varphi(C_2H_6)$ | 0.299351 |
| 6 | $C_2H_6$ | 0.066181 | $\varphi(CH_4)$ | 0.29588 |
| 7 | $\%CH_4$ | 0.082497 | $\varphi(H_2)$ | 0.292157 |
| 8 | $\varphi(C_2H_2)$ | 0.067574 | $CO_2/CO$ | 0.298659 |
| 9 | $CO_2$ | 0.066806 | $C_2H_2/H_2$ | 0.304623 |
| 10 | $C_2H_6/C_2H_2$ | 0.068373 | $C_2H_6/C_2H_2$ | 0.305679 |
| 11 | $C_2H_2$ | 0.059614 | $C_2H_2/CH_4$ | 0.30906 |
| 12 | $CH_4/H_2$ | 0.055447 | $C_2H_2/C_2H_4$ | 0.301607 |
| 13 | $C_2H_4$ | 0.057832 | $C_2H_4/C_2H_6$ | 0.306031 |
| 14 | $\varphi(CH_4)$ | 0.064196 | $CH_4/H_2$ | 0.301547 |
| 15 | $\%H_2$ | 0.032973 | $\%CH_4$ | 0.303705 |
| 16 | $\varphi(C_2H_6)$ | $-0.00899$ | $\%C_2H_2$ | 0.279383 |
| 17 | $CH_4$ | 0.002819 | $TH$ | 0.26508 |
| 18 | $CO$ | $-0.00451$ | $CO_2$ | 0.243314 |
| 19 | $C_2H_2/C_2H_4$ | $-0.01682$ | $CO$ | 0.245554 |
| 20 | $TH$ | $-0.00641$ | $C_2H_6$ | 0.246708 |
| 21 | $C_2H_2/H_2$ | $-0.02399$ | $C_2H_4$ | 0.229841 |
| 22 | $\%H_2$ | $-0.02539$ | $C_2H_2$ | 0.213115 |
| 23 | $\varphi(H_2)$ | $-0.0789$ | $CH_4$ | 0.202028 |
| 24 | $CO_2/CO$ | $-0.08773$ | $H_2$ | 0.187682 |

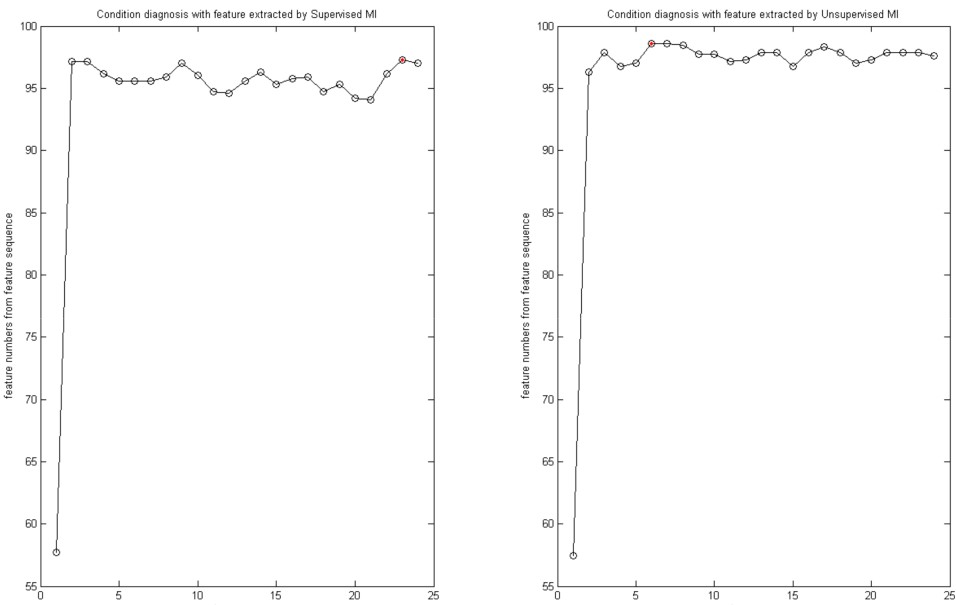

**Figure 4.** Diagnostic precision with supervised mutual information feature selection vs. unsupervised approach.

As can be seen from Figure 4, features selected from both unsupervised and supervised feature selection methods have good performance as input to diagnostic accuracy in cases, and both of the models achieve greatly increased diagnostic accuracy in the second feature.

### 5.3.2. Diagnostic Precision by Features of the Unsupervised Approach with Best Fitness vs. Other Classical Feature Set

Feature set with the highest diagnostic accuracy obtained by unsupervised mutual information feature selection method is used as input to optimized SVM diagnosis model for main transformer cases, other typical feature sets are used in contrast as shown in Table 4.

**Table 4.** Diagnostic precision by features of the unsupervised approach with best fitness vs. other classic feature set (percent).

| TYPE | 1 | 2 | 3 | 4 |
|---|---|---|---|---|
| 5-FOLD-1 | 100.00 | 89.29 | 89.29 | 96.43 |
| 5-FOLD-2 | 100.00 | 100.00 | 100.00 | 100.00 |
| 5-FOLD-3 | 100.00 | 96.43 | 96.43 | 100.00 |
| 5-FOLD-4 | 100.00 | 92.86 | 100.00 | 100.00 |
| 5-FOLD-5 | 92.86 | 96.43 | 92.86 | 92.86 |
| AVERAGE | 98.57 | 95.00 | 95.71 | 97.86 |

Note: 1 refers to the features of the unsupervised approach with the best fitness obtained as shown in Figure 4 as input to the optimized svm model; 2 refers to the features used in the three ratios method [1]; 3 refers to the features used in some intelligent methods [16]; 4 refers to the features used in the Duval Triangle method [2].

As shown in Table 4, the feature set obtained by the unsupervised mutual information feature selection algorithm, is used as the input of the optimized SVM diagnosis model and performs better than other inputs of the feature set in the case of diagnosis of the main transformer condition diagnosis. Therefore, the algorithm has high applicability.

### 6. Conclusions and Analysis

Fault data of main transformer lacks. In addition, the fault mode of the main transformer is different from that of other power transformers. This paper proposes an unsupervised mutual information feature selection method to calculate DGA monitoring data of main transformer and output feature selection sequence. Compared with the supervised mutual information feature selection algorithm, the unsupervised mutual information feature selection algorithm is highly correlated with the sequence features output by the supervised feature selection algorithm in feature selection. In the samples, the training samples and test samples were designed by five-fold method based on the appropriate feature set obtained by the unsupervised mutual information feature selection algorithm. The PSO optimized support vector machine model was used to verify the main transformer fault diagnostic, and the diagnosis accuracy was high. This method is suitable for feature extraction in main transformer fault diagnosis. However, the feature extraction method based on unsupervised mutual information is essentially an embedded feature extraction method with some significant advantages and disadvantages at the same time. The redundancy between features in the selected feature set is minimized, and its limitations depend on the evaluation of candidate solutions by the classification algorithm, which is computationally more expensive. Therefore, the offline data set can be used for training and verification in practical application, and the obtained feature set can be used to judge the condition of nuclear power transformers online.

**Supplementary Materials:** The following are available online at https://www.mdpi.com/article/10.3390/su14052700/s1, the experimental data of DGA samples used in Section 5.1 of this paper.

**Author Contributions:** Conceptualization, R.Y.; data collection, J.T.; methodology and the other, W.Y. All authors have read and agreed to the published version of the manuscript.

**Funding:** This research received no external funding.

**Institutional Review Board Statement:** Not applicable for studies not involving humans or animals.

**Informed Consent Statement:** Not applicable.

**Data Availability Statement:** Data can be obtained in the Supplementary Materials of the paper.

**Conflicts of Interest:** Jun Tao is an employer from CNNC Nuclear Power Operation Management Co., Ltd. The other authors declare that they have no conflict of interest.

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
