# Peer review of "An Unsupervised Mutual Information Feature Selection Method Based on SVM for Main Transformer Condition Diagnosis in Nuclear Power Plants"

_sustainability, doi:10.3390/su14052700_

Round 1
Reviewer 1 Report
This paper proposes an unsupervised mutual information feature selection method based on SVM for main transformer condition diagnosis in nuclear power plants. Although the main ideas are clearly expressed and simulation results of the proposed method are convincing, there are some aspects should be modified or explained to improve the overall quality of the manuscript.
- It is better to improve the quality of text, because there are some English errors and typos throughout the paper.
- In Page 4, column 3, part 3.2. The parameters of "PSO Settings" include the parameters of PSO and SVM. It is suggested that the author can use the "PSO-SVM Setting" as the title of this part.
- In this paper, some parameter symbols are unclear and inconsistent. Please clarify and unify the expression of parameters.
- In Page 5, column 4 , part 4.2. According to [12], formula (2) is incorrect. Rel(fi) should be the mean value of all mutual information values between individual feature fi and class c.
- The advantages of the proposed method are described in the conclusion of this paper. Moreover, the shortcomings of the proposed method or the future work can be added to the conclusion.
Author Response
1.It is better to improve the quality of text, because there are some English errors and typos throughout the paper.
Response 1: I would carefully examine the grammar and expression of the paper.
2.In Page 4, column 3, part 3.2. The parameters of "PSO Settings" include the parameters of PSO and SVM. It is suggested that the author can use the "PSO-SVM Setting" as the title of this part.
Response 2: Sure.
3. In this paper, some parameter symbols are unclear and inconsistent. Please clarify and unify the expression of parameters.
Response 3: I would carefully examine each parameter symbols to avoid any confusion or error.
4. In Page 5, column 4 , part 4.2. According to [12], formula (2) is incorrect. Rel(fi) should be the mean value of all mutual information values between individual feature fi and class c.
Response 4: I would like to discuss about the point. According to literature [12], formula (1) of this paper is proposed, and for the unsupervised data set, as you said, that is, the data set without label data, Relevance of feature fi is the average mutual information between feature f(i) and any other one in the whole Feature set is defined as Rel(fi), as denoted in Formula 2.
If this theory applies to labeled data sets, where c is the label of the data set, and the Relevance is the mutual information between that feature and the label set, as yellow labeled section 3.2.2 fomular(5) in the appendix and the relationship between these two methods is proved mathematically in [13].
5.The advantages of the proposed method are described in the conclusion of this paper. Moreover, the shortcomings of the proposed method or the future work can be added to the conclusion.
Response 5: I would refine the conclusion as you suggested.

Reviewer 2 Report
Dear Authors,
Thank you very much for the interesting paper.
The paper presents a study about unsupervised mutual information feature selection method, which is based on SVM for transformer diagnostics in nuclear power plants. Authors say that dissolved gas in oil, which is called DGA method, is a common means of monitoring the condition of transformer. The concentration of gas is important index to indicate the transformer condition. The special operation and maintenance modes are able to say the fault modes particularity of transformers. To solve the problem of insufficient samples, the paper puts unsupervised mutual information method to select the feature, which is verified by the optimized support vector machine, and tries to find the feature sequence with better performance.
Comments and question:
- Introduction chapter does not consist of information what defects of power transformers can be found by DGA method. I think, it could be important information for readers. Please complete.
- Chapter 5 – authors mentioned about thermal and electrical faults. There is no information what kind of thermal and electrical faults they mean. There are many types of thermal faults in power transformers, for example. But not all of them are recognized by the use of DGA method. Please complete.
- The same comment is for electrical faults. Please complete in chapter 5 or 1.
- Some information are expected in case of chapter 5, according to Duval triangle, where authors study gas level and their ratio. Please complete.
Author Response
1.Introduction chapter does not consist of information what defects of power transformers can be found by DGA method. I think, it could be important information for readers. Please complete.
Response 1: I would give some description about how DGA method applied to power transformer condition diagnosis.
2.Chapter 5 – authors mentioned about thermal and electrical faults. There is no information what kind of thermal and electrical faults they mean. There are many types of thermal faults in power transformers, for example. But not all of them are recognized by the use of DGA method. Please complete.
Response 2: I would give some explanation about how DGA method applied to classify different faults type similar to point 2.
3.The same comment is for electrical faults. Please complete in chapter 5 or 1.
Response 3: I would complete it as you suggest.
4.Some information are expected in case of chapter 5, according to Duval triangle, where authors study gas level and their ratio. Please complete.
Response 4: I would clarify the features applied in the case of chapter 5.